# Performance of Grating Couplers Used in the Optical Switch Configuration

**DOI:** 10.3390/s23229028

**Published:** 2023-11-07

**Authors:** Emilie Laffont, Arnaud Valour, Nicolas Crespo-Monteiro, Pierre Berini, Yves Jourlin

**Affiliations:** 1School of Electrical Engineering and Computer Science, University of Ottawa, Ottawa, ON K1N 6N5, Canada; pberini@uottawa.ca; 2Department of Physics, University of Ottawa, Ottawa, ON K1N 6N5, Canada; 3Université de Lyon, Laboratoire Hubert Curien, UMR CNRS 5516, 42000 Saint-Etienne, France; arnaud.valour@univ-st-etienne.fr (A.V.); nicolas.crespo.monteiro@univ-st-etienne.fr (N.C.-M.); yves.jourlin@univ-st-etienne.fr (Y.J.); 4Nexus for Quantum Technologies Institute, Advanced Research Complex, Ottawa, ON K1N 6N5, Canada

**Keywords:** surface plasmon resonance, plasmonic diffraction grating, optical switch, laser interference lithography, nanoimprinting

## Abstract

Surface plasmon resonance is an effect widely used for biosensing. Biosensors based on this effect operate in different configurations, including the use of diffraction gratings as couplers. Gratings are highly tunable and are easy to integrate into a fluidic system due to their planar configuration. We discuss the optimization of plasmonic grating couplers for use in a specific sensor configuration based on the optical switch. These gratings present a sinusoidal profile with a high depth/period ratio. Their interaction with a *p*-polarized light beam results in two significant diffracted orders (the 0th and the −1st), which enable differential measurements cancelling noise due to common fluctuations. The gratings are fabricated by combining laser interference lithography with nanoimprinting in a process that is aligned with the challenges of low-cost mass production. The effects of different grating parameters such as the period, depth and profile are theoretically and experimentally investigated.

## 1. Introduction

In recent years, surface plasmon resonance (SPR) biosensors were optimized and became competitive with gold standards such polymerase chain reaction (PCR) tests [1] and enzyme-linked immunosorbent assays (ELISA) [2]. Kretschmann and Raether were the first to propose a coupling method based on a prism to excite the SP in 1968 [3]. Nowadays, this coupling configuration is certainly the most widespread behind SPR sensing [4,5,6,7].

Yet, this configuration presents inconveniences such as a bulky arrangement hard to integrate into a fluidic system setup, which limits its miniaturization prospects. Recent studies highlighted [8,9] and demonstrated [10,11] the great potential of nanostructures to address this challenge. Especially, the great tunability and the integrability of plasmonic grating couplers into detection devices motivate the optimization of their designs to enhance the performance of sensors. Indeed, several studies highlighted the effects induced by different grating parameters on their optical response, such as the shape of their profile [12,13,14], their period [15,16] and their depth [17,18,19]. However, the optimization of configurations using a grating coupler was manifested through more elaborate designs [12,20,21], often resulting in cumbersome and expensive fabrication protocols which are not well aligned with low-cost mass manufacturing.

In 2014, Sauvage et al. reported for the first time a new plasmonic effect termed the “optical switch” [22] and proposed using this effect for anticounterfeiting applications. This effect involves a transfer of energy between two orders diffracted by a deep sinusoidal metallic grating, as the angle of incidence is varied over a few degrees about a specific angle of incidence, where the power in both diffracted orders is equal. Sensing based on this effect was demonstrated for the first time by detecting small changes in the bulk refractive index of aqueous [23] and gaseous media [24] with very good performance (in the range of 10^−6^–10^−7^ RIU) in terms of limit of detection (LOD). The main advantages of this configuration include the ability to perform differential measurements in real time using a simple interrogation setup and the ability to fabricate the grating couplers cost-effectively.

Indeed, a cost-effective and simple fabrication protocol based on laser interference lithography (LIL) [25] and nanoimprinting [26] was developed to produce the gratings. LIL is a cheap and fast technique of production for periodic nanostructures over large areas. It consists of transferring an optical interference pattern resulting from the overlap of several coherent laser beams onto a photoresist film. This method of production is also aligned with low-cost manufacturing and enables the production of various nanostructure profiles [25,27], including sinusoidal gratings. Yet, photoresist nanostructures are not sufficiently resilient (chemically and mechanically) to survive long exposures to fluids in biosensing applications, resulting in deformation of their surface.

Nanoimprinting enables the transfer of the grating pattern to a more resilient material than photoresist, such that samples are reusable many times, contrary to samples directly fabricated by LIL. Also, it enables the production of many similar samples from a single master and thus is especially appealing for mass production. Finally, nanoimprinting reduces the average roughness of the samples compared to the initial master. Roughness has a non-negligible effect on the optical response of gratings with a high depth/period ratio and should be minimized [28].

In this paper, we investigate theoretically and experimentally variations in structural parameters such as the period, depth and profile of grating couplers operating in the optical switch configuration, aiming to optimize performance. Section 2 describes the materials and fabrication methods adopted, Section 3 summarizes and discusses the experimental results relative to theory, and Section 4 gives brief conclusions.

## 2. Material and Methods

### 2.1. Production of Grating Masters

The first step in the realization of the gratings consisted of producing grating masters in a photoresist by LIL with depths greater than those expected for the final grating replicas to compensate for depth losses incurred during the nanoimprinting process (our nanoimprinting process is presented and discussed in detail in Section 2.2). The production of grating masters required three sequential steps as described below.

First, the adhesion promoter SurPass^TM^ 4000 (DisChem, Ridgway, CO, USA) was used to improve the adhesion of the photoresist on a 26 × 26 mm^2^ glass substrate. The adhesion promoter was applied via a static dispense and then spun in two sequential steps at 3000 rpm for 10 s and at 5000 rpm at 50 s. Next, this process was repeated with the positive photoresist Shipley S1828 (MICROPOSIT, Berlin, Germany). Finally, the samples were soft baked for 1 min at 60 °C to evaporate the solvent and increase the density of the photoresist layer. To prevent imperfections and discontinuities in the photoresist film, each glass substrate was initially meticulously cleaned via three sequential steps: ultrasonic cleaning in acetone for 10 min, ultrasonic cleaning in ethanol for 10 min, and static incubation in de-ionized water (DI) for 10 min. Finally, the samples were dried under nitrogen gas.

LIL is an inexpensive lithography technique to produce gratings over large areas. This method of exposure is aligned with low-cost manufacturing and enables the production of various grating profiles [25,27], including sinusoidal gratings. The photoresist-coated samples were first exposed to a uniform He-Cd laser beam at a wavelength of λ = 442 nm and an incident optical power density of 204 μW/cm^2^ for 115 s to reach the linear operating regime of the photoresist. A second exposure was then applied, consisting of the superposition of two balanced and coherent beams from the same laser at an incident power density of 408 μW/cm^2^ for 115 s, to produce the desired interferometric sinusoidal profile. During this second exposure, the grating period, Λ, in the photoresist was fixed to 770 nm by the laser wavelength, λ, and the angle of incidence of the laser beams, θ (controlled by a Labview program), based on the relation below:$$\Lambda =\frac{\lambda }{2}\;sin(\theta ).$$

The interference gave rise to a fringe pattern due to the overlap of both beams. This period was chosen based on theoretical optimization of the switch pattern, as discussed in Section 3.

Finally, the samples were developed in the developer MF319 (MICROPOSIT) for 9 s. Thus, 770 nm period gratings with a depth about 290 nm and an average roughness of 3–6 nm were obtained, as shown in Figure 1 (top row).

### 2.2. Production of Grating Replicas

In previous work [23], we noticed that a prolonged exposure to fluids damaged the gratings produced by these steps in the S1828 photoresist, resulting in non-repeatable sensing measurements. A nanoimprinting process was thus developed to replicate grating masters in a more resilient material (Amonil MMS1, AMO). The sequential steps of the nanoimprinting process applied to produce grating replicas are described below.

A PDMS (polydimethylsiloxane) stamp was fabricated by replicating the sinusoidal grating master in S1828 via the following steps. First, a “hard” PDMS solution was formed by mixing in a 3:1 ratio (by weight) RTV615 silicone rubber compound with a curing agent (Momentive^TM^). The solution was then deposited on the master fabricated by LIL in S1828 by spin coating (10 s at 3000 rpm followed by 50 s at 5000 rpm). This deposition process fills the pattern of the master. The sample was then sequentially soft baked at 50 °C for 30 min and 70 °C for 1 h. Next, the hard PDMS coated master was placed in a sample holder and coated by a “soft” PDMS solution formed by mixing in a 9:1 ratio (by weight) the RTV615 silicon rubber compound with a curing agent. The sample was then soft baked at 70 °C for 2 h. Coating with a soft PDMS layer facilitates the removal and manipulation of the hard PDMS stamp from the master.

Two thin layers of Amoprime (AMO) and Amonil MMS1 were used as the adhesion promoter and the imprint resist into which grating replicas were transferred using the hard PDMS stamp. These layers were sequentially deposited via a static dispense and then spun in two sequential steps at 3000 rpm for 10 s and at 5000 rpm at 50 s on a clean 26 × 26 mm^2^ glass substrate. Then, the PDMS stamp was placed in contact with the sample in a printing press (REV 3S, Transmatic, Manchester, UK), and a low-imprint pressure of 1 Bar was applied for 30 s. The sample was then illuminated by a UV lamp (Ucube 365-100-2, Uwave) to harden the imprint resist. Finally, the stamp was released from the replica, which was then soft cured at 60 °C for 1 min. Thus, grating replicas with a period of 770 nm, a depth of 240 nm and an average roughness of 0.4–1 nm were obtained, as shown in Figure 1 (middle row). Figure A1 shows a partial AFM characterization of a corrugated grating replica obtained by applying this protocol.

### 2.3. Deposition of Thin Metal Layer

To form the final grating couplers, chromium and gold layers were sequentially deposited on the Amonil replicas by thermal evaporation. A thin chromium layer was used as an adhesion layer for the gold layer. The deposition rates and the final thicknesses were 0.3–0.4 nm/s and 6–7 nm for the chromium film and 1.3–1.4 nm/s and 120–121 nm for the gold layer, respectively. These two layers were deposited in a vacuum chamber pumped to a pressure of about 10^−6^ mbar. The thickness of the gold layer was chosen to be greater than 100 nm to prevent transmitted orders from emerging during the measurements.

### 2.4. Interrogation Setup

Metallized grating masters and replicas, such as those shown in Figure 2a,b, produced via the protocol detailed in the previous subsection, were placed in the setup sketched in Figure 2c for testing [23]. A collimated beam from a laser diode emitting at the free-space wavelength of λ_0_ = 850 nm and polarized TM (transverse magnetic) by an IR-polarizer was incident on the grating placed inside a flow cell, and the angle of incidence was varied in steps of a single degree. The angle of incidence was controlled using a rotation stage holding the cell and aligned along the central rotation axis using an xy stage. The flow cell was filled with de-ionized water (DI) used as reference solution for measuring the optical responses (switch patterns) of the grating interrogated over an angular scan. An aperture removed the background light and limited the diameter of the collimated beam from the laser diode to 1 mm. Figure A2 (Appendix A) shows that the normalized switch patterns do not depend on the diameter of the incoming beam. Two 5 × 5 mm^2^ Si-based photodiodes connected by tracks to rotation stages collected the power in the 0th and −1st orders. The rotation axes of both rotation stages were aligned along that of the rotational stage controlling the position of the flow cell. The photodiode currents were converted to voltages using transimpedance amplifiers giving an output signal (voltage) proportional to the incident optical signal. Labview software (https://mcgrating.com/ (accessed on 4 September 2023)) was used to perform data acquisition from both photodiodes.

### 2.5. Computational Method

The theoretical responses of the grating couplers presented in Section 3 were computed by modeling a grating embedded in water and probed by a *p*-polarized 850 nm light source over an angular range via the software MC-Grating (https://mcgrating.com/ (accessed on 4 September 2023)) based on Chandezon’s method [29,30]. The sample period was modeled by a linear combination of 45 sinusoids approximating one of the grating periods extracted from an AFM characterization. This period was considered as infinitely repeated to represent the fabricated grating. Table 1 gives the numerical parameters of the structure used to model the grating represented in Figure 3. The modeled sample consists of a quasi-sinusoidal (linear combination of 45 sinusoids) grating covered by a 100 nm thick gold layer bounded by water. This gold thickness was chosen to prevent transmitted orders. Thus, the refractive index of layer 3 (Table 1) does not have a significant effect on the computed results, since the gold layer can be considered as infinite. The Snell–Descartes law was applied to the incident angles in water to obtain their values in air and compared with the optical responses experimentally measured in air. For each simulation curve given in Section 3, 100 points were considered per curve.

## 3. Results

The effects of the grating parameters on SPR responses were widely discussed in the literature [14,15,16,19,31]. Here, we carry out studies to optimize the response of sinusoidal gratings with a higher depth/period ratio for use in the optical switch configuration. A switch pattern (optical response) is considered as optimized when the lateral minimal power of the −1st order at θ_+1_ and θ_−2_ as well as the central minimal power of the 0th order at Littrow’s angle correspond to the extinction of these orders, as shown in Figure 4. In addition, the angular distance between both operating angles θ_l_ and θ_r_ must be large enough to distinguish and reach them in an interrogation setup. This section investigates the effects of the period, the depth, and the profile on the optical response of gratings used in the switch configuration.

### 3.1. Effect of the Period

The pattern (angular response) of the optical switch can be modified by changing the period of the grating. Indeed, this parameter modifies the angular distance between both working points at θ_l_ and θ_r_ and the slopes associated with both diffracted orders in the linear region around these angles of incidence, which are highlighted by red boxes in Figure 4. The angular distance between both working points must be large enough to ensure that the photodiode collecting the −1st diffracted order does not obstruct the optical pathway of the probing beam from the laser diode when the angle of incidence is fixed at one of the working points in the arrangement shown in Figure 2c. Regarding the slopes in the linear region around each working point, their value determines the bulk and surface sensitivities of the system: higher slopes lead to higher sensitivity.

Figure 5a shows three computed switch patterns resulting from the interaction of a *p*-polarized light beam with sinusoidal gratings of different periods (Λ), as sketched in Figure 5b. These three structures theoretically provide an optimal switch pattern with extinctions of the −1st order at θ_+1_ and θ_−2_ as well as extinction of the 0th order at Littrow’s angle. The main differences between the switch patterns of these gratings are their slopes in the linear regions around each working point, as listed in Table A1 of the Appendix A, and the angular distance between these two points as highlighted by the double arrows in Figure 5a. Indeed, the 750 nm period grating (dark blue curves) presents a larger angular distance between both working points (black dashed cursors) than the other gratings. Its working points should be easier to access experimentally and to distinguish if one considers the optical setup comprising two photodiodes and a laser source. However, the 790 nm period grating (yellow curves) presents slightly higher slopes (Table A1, Appendix A) in the linear region around each working point and thus a better sensitivity than the other gratings. The 770 nm period grating (bright blue curves) yields a trade-off between the other two gratings.

Figure 6a shows the normalized experimental and theoretical switch patterns of a 790 nm period grating as sketched in Figure 6b, which was produced by LIL following the protocol detailed in Section 2. This structure was fabricated directly in S1828 photoresist and was not replicated, thereby providing baseline performance characteristics of a master. Figure 6c shows an AFM profile of the fabricated grating. The theoretical switch patterns were computed by approximating the periods shown in the color boxes of Figure 6c as a linear combination of 45 sinusoids and repeating them infinitely to represent the fabricated grating.

Both experimental switch patterns (light blue and orange curves) were measured at different locations on the same sample, and they are very similar. This suggests good uniformity of the grating over the sample. However, they are not as optimal as the theoretical switch patterns (dark blue and yellow curves) simulated using the periods extracted from the AFM profile as described above. Indeed, the experimental working points are rather close to Littrow’s angle, θ_L_, and also hard to reach without interfering with the incident beam. Thus, the measurements of the −1st order were only performed over narrow angular scans between each working point and Littrow’s angle, as shown in Figure 6a. In addition, the contrast in power between the 0th and the −1st orders at Littrow’s angle is not as high as expected compared to the theoretical switch patterns. We assume that these differences between the theoretical and experimental patterns are in part due to roughness on the Au film and to the distorted profile of the grating. Indeed, grating roughness can modify significantly the switch pattern, as reported in ref. [28].

Regarding the roughness effect, this hypothesis is in part supported by the theoretical switch patterns given in Figure 6a, which were simulated by considering different regions of the AFM scan as shown in Figure 6c. The period in the dark blue box is rougher than that in the yellow box, yielding a theoretical switch pattern of lower contrast (dark blue curves) between the −1st and the 0th orders at Littrow’s angle, than the smoother period (yellow curves).

We observed that the profile of the grating became modified and damaged after a prolonged exposure to fluids encountered in biosensing, which compromised the reliability and the repeatability of the measurements and could partially explain the mismatch between the theoretical and experimental switch patterns. This observation motivated the replication of the gratings in Amonil, as described in Section 2.

To ensure a large enough angular distance between both working points, gratings of a smaller period (770 nm) were produced. Figure 7a shows the normalized experimental (light blue and orange curves) and theoretical (dark blue and yellow curves) switch patterns of this grating, which was produced by replication in Amonil (Figure 7b). Similarly to the 790 nm period grating, the experimental switch patterns measured at two different locations on the grating surface are very similar. But the working points are easier to reach, as shown by measurements acquired over broader angular scans between both working points and Littrow’s angle, θ_L_, compared to the 790 nm period grating.

In addition, the experimental switch patterns agree quite well with the theoretical ones, which are themselves more similar to each other and closer to optimal than those of the 790 nm period grating. This suggests that the 770 nm period grating is smoother and more regular than that with a period of 790 nm, primarily due to the improved quality of the replicated gratings, which are more resilient and smoother, as shown in Figure 1. Indeed, as mentioned in Section 2, the nanoimprinting process provided grating replicas with a lower average roughness than their initial grating master. We assume that the decrease in the average roughness of about a few nanometers between the replicas and their master is due to the high enough surface tension of amonil, which results in losses in depth and the removal of slight roughness of the initial master. Figure A3 gives the distribution of replica depths from two grating masters with an initial depth, d, of 285 nm (green) and 337 nm (blue).

Part of the depth loss occurs when the sinusoidal pattern is transferred from the PDMS stamp to the grating replica, because the replicas produced from the same PDMS stamp do not have the same depth, as shown in Figure A3. Indeed, although most of the replicas have a depth of about 250–260 nm, some of them present higher and lower depths. Part of the depth loss is also likely due to the transfer of the initial sinusoidal pattern defined in photoresist by LIL to the hard PDMS stamp. That would explain why no replica reached the same depth as the initial grating master.

Also, theoretically, the 790 nm grating should produce larger slopes (Table A1, Appendix A) around the working points than the 770 nm grating (Figure 5a). However, this was not observed experimentally, as can be appreciated by comparing Figure 6a and Figure 7a (cf., Table A2 and Table A3, Appendix A) due to the degraded quality of the former after exposure to fluids for each switch pattern presented in Figure 5a and Figure 6a.

### 3.2. Effect of the Depth

The main difference between a grating operating in the optical switch configuration and a conventional grating coupler is the ratio of the depth to the period of the grating. In the case of the optical switch, this ratio is higher because the depth of the grating must be high enough to enable inter-plasmonic coupling between the backward and the forward plasmonic modes [22].

Figure 8 shows how increasing the grating depth modifies the optical response of a 100 nm thick gold-coated sinusoidal grating in water. The panels illustrate the evolution of a conventional grating coupler (top left) into a grating operating as an optical switch (bottom right). As shown in this figure, the depth has a strong effect on the contrast between the diffraction efficiencies of the −1st and the 0th orders, especially at Littrow’s angle, θ_L_. The sensitivity and the dynamic range of a detection device working in the optical switch configuration are affected by the contrast between both diffracted orders at Littrow’s angle. Thus, the depth must be rigorously chosen to optimize the switch pattern and the performance of the system.

Figure 9 shows three normalized experimental switch patterns measured on three different replicas produced from the same grating master. The gratings have identical profiles and the same period, but their depth varies slightly due to variations in the depth losses due to the nanoimprinting process used to produce the replicas (Figure A3). Figure 8 and Figure 9 show that beyond a threshold depth, the switch pattern is no longer optimal. Indeed, the switch pattern measured on the 244 nm deep grating (dark blue curves) presents a good contrast between the normalized power of the −1st and 0th orders at Litttrow’s angle, θ_L_. But the deeper grating replicas (252 and 279 nm depth) present switch patterns of lower contrast at this angle. A similar effect is observed by comparing the switch patterns computed for the 220 and 260 nm deep gratings. This demonstrates that despite the large ratio between the depth and the period of the gratings used to produce a switch pattern, the depth of the gratings must not be above a threshold value, or the switch pattern degrades. These experimental observations were supported by theory, as shown in Figure A4, which gives the computed power distribution in the 0th and the −1st orders resulting from the illumination of a sinusoidal gold-coated grating bounded by water (sensing medium) as the angle of incidence of the probing beam is varied.

### 3.3. Effect of the Profile

Contrary to the period and the depth, the profile of the grating is harder to control. Yet, its effect on the switch pattern is very critical. Figure 10a shows three experimental switch patterns (light blue, orange and yellow curves) and a simulated one (dark blue curves). The three experimental switch patterns were obtained on three different sinusoidal gratings replicated from different masters by two distinguished processes. The light blue and orange switch patterns were measured on grating replicas produced based on the entire protocol detailed in Section 2, while the yellow switch pattern was measured on a grating replicated by another NIL equipment from a grating master produced by the LIL protocol described in Section 2. The three grating replicas had a period 770 nm and similar depths (light blue curves: 228 nm; orange and yellow curves: 233 nm). They were coated with 121 nm of gold in the same deposition run. The theoretical switch pattern was simulated based on a perfect sinusoidal grating of the same period (770 nm) and an intermediate depth (230 nm). Figure 10b–d compare a perfect sinusoid with one period extracted from an AFM scan of each grating used in the measurements given in Figure 10a.

Despite having the same period and very similar depths, the grating with the period shown in Figure 10b yields a switch pattern (light blue) in better agreement with the simulation than the gratings with the periods given in Figure 10c,d (orange and yellow switch patterns, respectively). We assume that this difference in the optical responses is due to the slightly different profile of these gratings. Indeed, the period shown in Figure 10b agrees better with a perfect sinusoid than those shown in Figure 10c,d. Moreover, the period shown in Figure 10c presents a better overlap with a perfect sinusoid than that given in Figure 10d, which is also rougher than the periods shown in Figure 10b,c. The poor profile of this grating (Figure 10d) explains its poor switch pattern, which is plotted as the yellow curves in Figure 10a. This phenomenon was observed for other gratings of different depths, as shown in Figure A5. Thus, the profile of gratings used in an optical switch configuration must be smooth and of high quality in order for the switch pattern to be optimal.

## 4. Conclusions

The effects of design parameters such as the period, depth and profile of deep gratings used as couplers were theoretically and experimentally investigated for the case where the grating is used in the optical switch configuration. The better agreement between the theoretical and experimental responses observed for the grating replicas demonstrates the interest in combining LIL with nanoimprinting as a fabrication process. Indeed, the prolonged exposure of grating masters (directly produced by LIL) to fluids distorted the sinusoidal profile of these structures. Yet, the effect of the grating profile on their optical response, especially on the angular distance between the working points and the contrast between the 0th and the −1st orders, is very strong, which in turn affects the performance (sensitivity) in sensing. In addition, the grating replicas produced by nanoimprinting presented a lower average roughness than the grating master directly produced by LIL and provided better switch patterns. Thus, an optimized design of grating replicas was fabricated with a period of 770 nm, a depth of 228 nm, and an average roughness of 0.4–1 nm. Moreover, the combination of LIL and nanoimprinting is cost-effective and suitable for volume production, and it provides many reusable samples from each initial grating master. These fabrication advantages combined with the competitiveness of SPR sensing using the optical switch configuration hold promise for point-of-care detection applications.

## Figures and Tables

**Figure 1 sensors-23-09028-f001:**
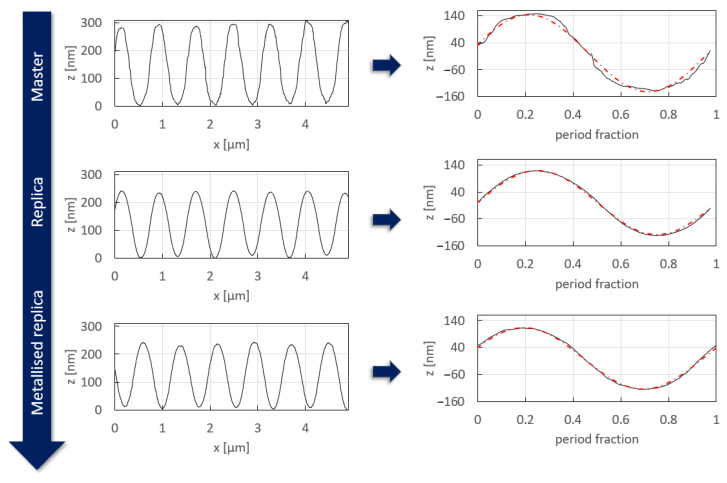
Evolution of the grating profile from the initial master (top row) to the metallized grating replica (bottom row) as measured by AFM scans (left panels). The right panels show a period extracted from each AFM profile with a perfect sinusoidal fit (red dashed–dotted curve).

**Figure 2 sensors-23-09028-f002:**
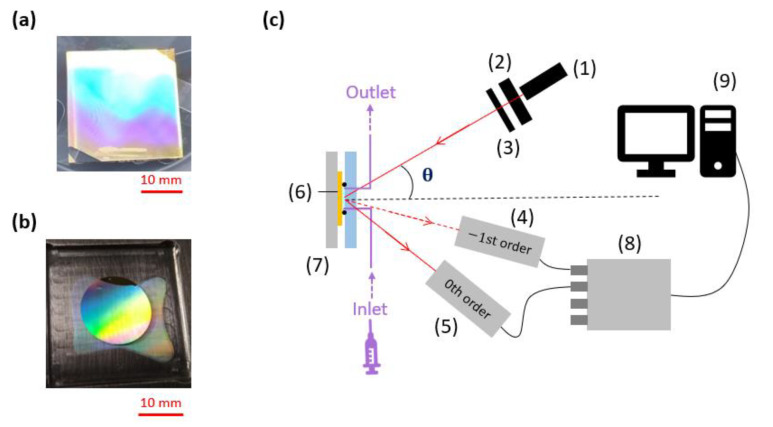
(**a**) Picture of a gold-coated master grating on a 26 × 26 mm^2^ glass slide. (**b**) Picture of a gold-coated grating replica on a 26 × 26 mm^2^ glass slide. (**c**) Setup used to perform the measurements, comprising an 850 nm wavelength laser diode (1), a polarizer (2), an aperture (3), and two photodiodes (4) and (5) to measure the power in the −1st (dashed red line) and the 0th (solid red line) orders diffracted from the gold-coated grating (6) placed within a fluidic cell (7) into which fluids are injected via peek tubing interfaces (purple) connected to a syringe pump. A DAQ (data acquisition) device (8) and a computer (9) were used to record the measurements. This figure is extracted from [23]. Reprinted with permission from https://www.mdpi.com/1424-8220/23/3/1188 (accessed on 4 September 2023). Copyright©2023 by the authors. Licensee MDPI, Basel, Switzerland. The Authors, licensed under a Creative Commons Attribution (CC BY) license.

**Figure 3 sensors-23-09028-f003:**
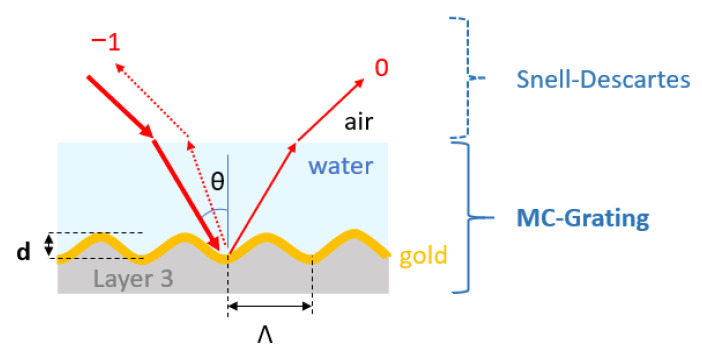
Structure used to model the interaction of a *p*-polarized light beam emitting in 850 nm at free-space wavelength and a grating with period Λ, depth d.

**Figure 4 sensors-23-09028-f004:**
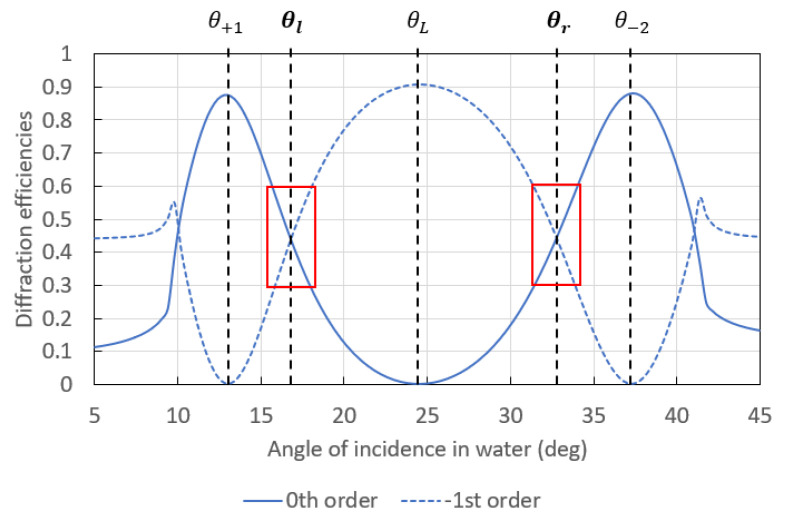
Computed switch pattern resulting from the superposition of the angular spectra associated with the −1st (dashed curve) and the 0th (solid curve) diffraction orders for a sinusoidal gold grating with a period of 770 nm and a depth of 220 nm probed in water by an incident *p*-polarized light beam at free-space wavelength λ_0_ = 850 nm. This switch pattern was computed via MC-Grating software. Both red boxes delimit the linear region of each working point.

**Figure 5 sensors-23-09028-f005:**
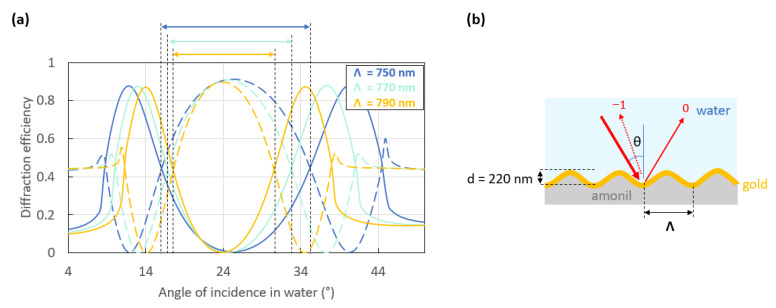
(**a**) Switch pattern resulting from the interaction of a *p*-polarized beam with the sinusoidal grating sketched in (**b**). This 100 nm gold-coated grating is interrogated at an incident free-space wavelength of 850 nm over an angular scan in water. The period is the only parameter varied: Λ = 750 nm (dark blue curves), Λ = 770 nm (bright blue curves), and Λ = 790 nm (yellow curves). The dashed curves represent the −1st order while the solid curves represent the 0th order. The double arrows correspond to the angular distance between both working points for each switch pattern.

**Figure 6 sensors-23-09028-f006:**
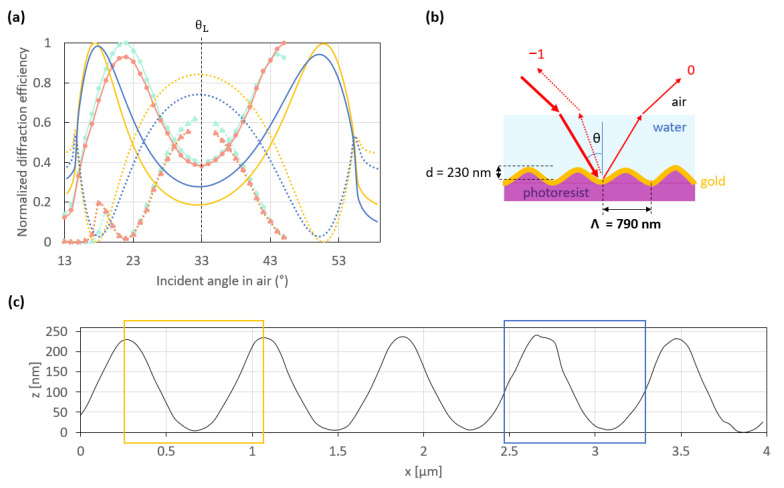
(**a**) Normalized theoretical (dark blue and yellow curves) and experimental (light blue and orange curves) switch patterns resulting from the interaction of an incident *p*-polarized beam at a free-space wavelength of 850 nm with the structure illustrated in (**b**). The dotted curves represent the −1st order, while the solid curves represent the 0th order. The theoretical switch patterns were simulated considering the period highlighted with boxes of the same color in (**c**) which gives the AFM profile of the 100 nm thick gold-coated grating used to measure the experimental switch pattern in (**a**). The experimental switch patterns were normalized by dividing the respective measurements (photodiode output voltage of the transimpedance circuit) by the maximum value. The theoretical switch patterns were normalized in the same way as the experimental ones (using the computed diffracted waves) to be directly compared.

**Figure 7 sensors-23-09028-f007:**
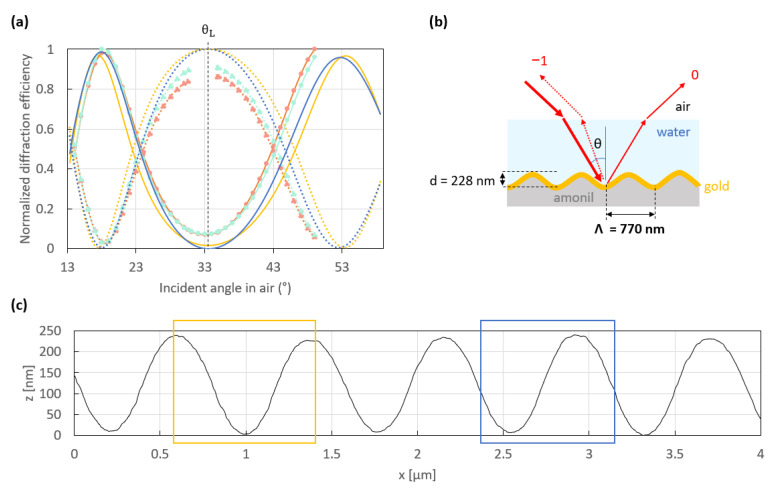
(**a**) Normalized theoretical (dark blue and yellow curves) and experimental (light blue and orange curves) switch patterns resulting from the interaction of an incident *p*-polarized beam at a free-space wavelength λ_0_ = 850 nm with the structure sketched in (**b**). The dotted curves represent the −1st order, while the solid curves represent the 0th order. The theoretical switch patterns were simulated considering the period highlighted with boxes of the same color in (**c**) which gives the AFM profile of the 121 nm thick gold-coated grating used to obtain the experimental switch patterns in (**a**). The experimental switch patterns were normalized by dividing the respective measurements (photodiode output voltage of the transimpedance circuit) by the maximum value. The theoretical switch patterns were normalized in the same way as the experimental ones (using the computed diffracted waves) to be directly compared.

**Figure 8 sensors-23-09028-f008:**
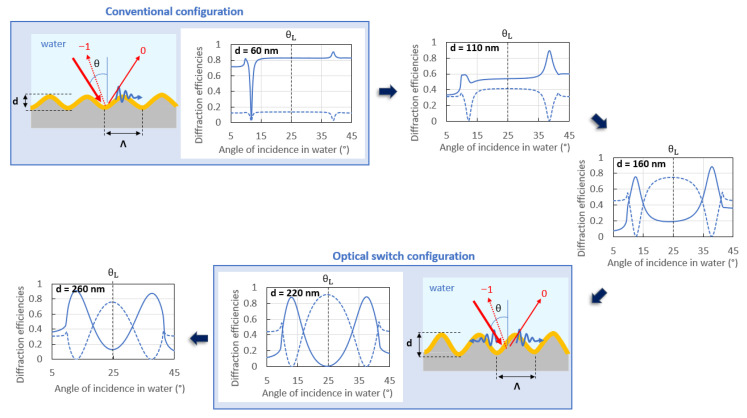
Normalized theoretical responses illustrating the effects of the grating depth on the optical response in water of sinusoidal gratings interacting with a *p*-polarized incident beam at a free-space wavelength λ_0_ = 850 nm. The panels illustrate the evolution of a conventional grating coupler (top left) into a grating operating as an optical switch (bottom right) as the depth increases.

**Figure 9 sensors-23-09028-f009:**
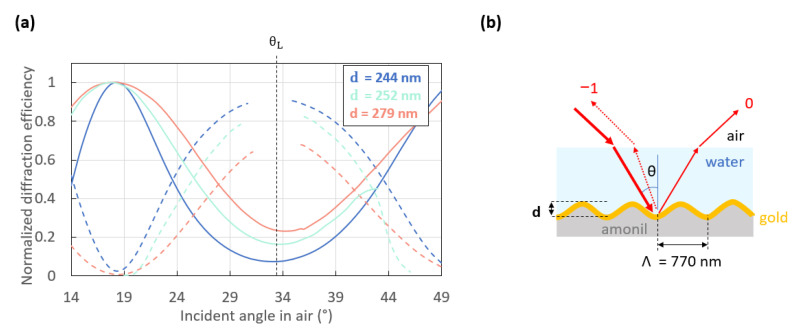
(**a**) Normalized experimental switch patterns resulting from the interaction of an incident *p*-polarized beam at a free-space wavelength λ_0_ = 850 nm with the structure sketched in (**b**).

**Figure 10 sensors-23-09028-f010:**
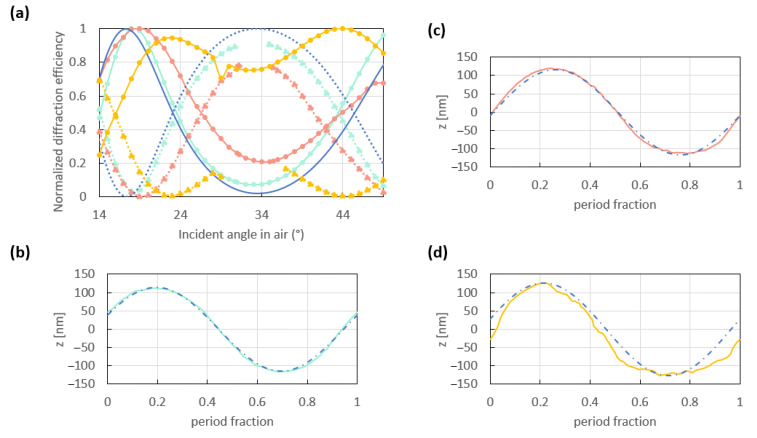
(**a**) Normalized theoretical (dark blue curves) and experimental (light blue, orange and yellow curves) switch patterns resulting from the interaction of an incident *p*-polarized beam at a free-space wavelength of λ_0_ = 850 nm with sinusoidal 121 nm thick gold-coated gratings of different profile. The theoretical switch pattern was computed for a perfect 230 nm deep and 770 nm period sinusoid. The three gratings used to measure these optical switch patterns had a 770 nm period and a similar depth of 228 nm (light blue) and 233 nm (orange and yellow). One period extracted from the AFM measurements of each grating is compared to a perfect sinusoid in (**b**–**d**), corresponding to the light blue, orange and yellow switch patterns of (**a**), respectively.

**Table 1 sensors-23-09028-t001:** Numerical parameters used to model the grating with the software MC-Grating. Water and gold refractive indices were extracted from the MC-Grating table. Photoresist and amonil refractive indices were extracted from their manufacturer datasheet.

Layer Number (Thickness)	Layer Material	Refractive Index (λ_0_ = 850 nm)
1 (superstrate)	Water	1.3282
2 (100 nm)	Gold	0.17 + 5.3 i
3 (substrate)	Photoresist	1.625
3 (substrate)	Amonil	1.52

## Data Availability

All data underlying the results of the paper are present in the paper.

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
