# Peer review of "Performance of Grating Couplers Used in the Optical Switch Configuration"

_sensors, 2023, doi:10.3390/s23229028_

Round 1
Reviewer 1 Report
Comments and Suggestions for Authors
In this paper the authors investigate configurations for plasmonic grating couplers to improve their performance in the 'optical switch configuration'. This investigation includes extensive numerical modeling, fabrication and experimental characterization. In general the paper is well written and the results are presented clearly. The authors show that by optimizing the depth of the grating it is possible to obtain better conditions for the plasmonic switch configuration, and also consider the effects of surface roughness and non-ideal grating profile.
One criticism that I do have for this paper is that while it is ostensibly aimed at improving the performance of the gratings when used in a 'specific sensor configuration' there is no quantitative assessment of the impact on sensing performance. Instead these aspects of the performance are treated qualitatively rather than quantitatively. In particular, the slope of the diffraction efficiency versus angle (in the linear regime) is cited as an important metric for sensor performance, and is used to justify why one depth may be better than another, but its value is not calculated. I suggest that the authors calculate the slope for each grating (and perhaps present it in a table), so that the different configurations can be compared more rigorously, and also so that the experimental and numerical results can be better compared. This would also be of benefit to other researchers who might want to follow up on this work.
A related question which would be interesting to address is the trade-off between slope and dynamic range for sensing (which I presume must exist).
Another parameter which seems to be important but which is again treated in a qualitative fashion is the angular separation between the left and right operating angles. We are informed that a larger angle is better, but no arguments are presented as to what the minimum practical angle is, or how this might impact performance.
Other more minor comments:
So far as I can see, reference 18 does not propose a sensing configuration based on the optical switch design.
What was the polarisation purity of the experimental laser beam?
On line 213 it is stated that "the angular distance between both operating angles theta_l and theta_r must be large enough to distinguish and reach them in an interrogation setup'. What does 'reach them' mean?
Related to this, in the authors' previous publication the two angles theta_l and theta_r, when used to label a figure, were typeset in italic font, whereas here they are typeset in roman font. As a result I initially read theta_l as theta_1, and thought it referred to a 1st order diffraction. I suggest going back to the italic font to reduce confusion.
Author Response
2023-10-23
Dear Mr. Ampansang,
Thank you for handling our manuscript and seeking reviewer reports. We are grateful to the reviewers for studying our manuscript and providing useful feedback. We have considered all reviewer comments, made several edits to our manuscript following suggestions, and respond to each in what follows.
Please do not hesitate to contact us if anything remains amiss with our manuscript.
Yours Truly,
Émilie Laffont
On behalf of all authors
Reviewer 1
- I suggest that the authors calculate the slope for each grating (and perhaps present it in a table), so that the different configurations can be compared more rigorously, and also so that the experimental and numerical results can be better compared.
We took into account this comment by adding three tables mentioning the slopes of the switch patterns presented in Figures 5(a), 6(a) and 7(a) in the Appendix section.
- A related question which would be interesting to address is the trade-off between slope and dynamic range for sensing (which I presume must exist).
The optical switch configuration using the grating coupler presented in this paper is used to detect small refractive index variations, so higher slopes are preferred to a larger dynamic range. A trade-off between sensitivity and dynamic range likely exists, and optimising the trade off is an interesting question that could be pursued in future work.
- Another parameter which seems to be important but which is again treated in a qualitative fashion is the angular separation between the left and right operating angles. We are informed that a larger angle is better, but no arguments are presented as to what the minimum practical angle is, or how this might impact performance.
A large enough angular distance between both working points implies that each working point is far enough from Littrow’s angle. That prevents the obstruction of the probing beam by the photodiodes (collecting the 0th and the -1st orders) in front of the laser diode due to the arrangement of the optical switch configuration presented in Figure 2(c). Thus, we ensure that no item of the setup is in the optical pathway of the probing beam before its interaction with the grating coupler. The new sentence added (line 213) to the manuscript should clear up the confusion.
- So far as I can see, reference 18 does not propose a sensing configuration based on the optical switch design.
Correct. Ref 18 describes the optical switch effect and demonstrates experimentally this effect based on spectral interrogation for security applications. We edited the text of our manuscript to clarify.
- What was the polarisation purity of the experimental laser beam?
As mentioned in our manuscript, “A collimated beam from a laser diode emitting at the free-space wavelength of λ0 = 850 nm and” was “polarized TM (transverse magnetic) by an IR-polarizer”. The TE-component was removed from the probing beam by a polarizer.
- On line 213 it is stated that "the angular distance between both operating angles theta_l and theta_r must be large enough to distinguish and reach them in an interrogation setup'. What does 'reach them' mean?
That means that we need working points far enough from Littrow’s angle to avoid the presence of the photodiode collecting -1st diffracted order in the optical pathway of the probing beam when it is fixed at one of the working points. Indeed, due to the arrangement of the optical switch configuration shown in Figure 2(c), if working points (θr and θl) are too close to Littrow’s angle, the photodiode collecting -1st diffracted order can obstruct the optical pathway of the probing beam from the laser diode. We edited our manuscript to clarify these points.
- Related to this, in the authors' previous publication the two angles theta_l and theta_r, when used to label a figure, were typeset in italic font, whereas here they are typeset in roman font. As a result I initially read theta_l as theta_1, and thought it referred to a 1st order diffraction. I suggest going back to the italic font to reduce confusion.
We changed the formatting of the angles mentioned in Figure 4, as suggested to avoid confusion.

Reviewer 2 Report
Comments and Suggestions for Authors
This study discusses the design and development of plasmonic grating couplers in a specific sensor configuration based on the optical switch. Numerical and experimental analyses were utilized to extract the spectral properties and show the promise of the proposed approach. Although the work has been prepared in a comprehensive style, there are some points that need to be addressed before proceeding further. Here are my comments and suggestions:
1) The first paragraph of the bibliography should be improved by adding more information about the recent state of the art approaches using plasmonic and photonic sensorics. See: Biosensors & Bioelectronics: X, 11, 100175 (2022) Laser & Photonics Reviews 16(2), 2100328 (2022), etc.
2) It seems that Figure 2c has been borrowed from another source. Instead, the authors should plot their own and use an original graph. Also, Figures 1a and 1b need proper scale bars.
3) A cross-sectional SEM image of the fabricated grating would be useful to assess the quality and precision of the fabrication method.
4) The precision of the sensor is not clearly obvious. The authors should provide a quantitative evaluation between the proposed method and the available ones from the literature.
5) Given that numerical studies were conducted to analyze the spectral properties of the design, the E-field maps and supplemental information should be provided.
Author Response
2023-10-23
Dear Mr. Ampansang,
Thank you for handling our manuscript and seeking reviewer reports. We are grateful to the reviewers for studying our manuscript and providing useful feedback. We have considered all reviewer comments, made several edits to our manuscript following suggestions, and respond to each in what follows.
Please do not hesitate to contact us if anything remains amiss with our manuscript.
Yours Truly,
Émilie Laffont
On behalf of all authors
Reviewer 2
- The first paragraph of the bibliography should be improved by adding more information about the recent state of the art approaches using plasmonic and photonic sensorics. See: Biosensors & Bioelectronics: X, 11, 100175 (2022) Laser & Photonics Reviews 16(2), 2100328 (2022), etc.
We added a few supplementary references about the great interest in the nanostructures to address the issues relative to the cumbersome arrangement of the classical Kretschmann’s configuration and their use in smartphone-based SPR biosensors.
- It seems that Figure 2c has been borrowed from another source. Instead, the authors should plot their own and use an original graph. Also, Figures 1a and 1b need proper scale bars.
This Figure was created by ourselves but we have already presented it in a previous article (reference [19]). That is why it is mentioned as extracted from (our) an article. Regarding the scale bars you suggested, we added them.
- A cross-sectional SEM image of the fabricated grating would be useful to assess the quality and precision of the fabrication method.
We do not have a SEM image of the grating profile, but we added an AFM profile of a grating fabricated by applying the protocol described in Section 2 as Figure A1 in Appendix section.
- The precision of the sensor is not clearly obvious. The authors should provide a quantitative evaluation between the proposed method and the available ones from the literature.
The limit of detection of this sensor was assessed in gazeous (10-7 RIU) and aqueous (10-6 RIU) sensing media as reported in two references [19,20] mentioned in the introduction. Reference [19] mentions similar studies reporting similar methods of detection in the literature. The present manuscript focuses on improving the response of the grating coupler used in the optical switch effect by investigating the effect of this grating design on its optical response. However, we edited our manuscript to include the LODs obtained.
- Given that numerical studies were conducted to analyze the spectral properties of the design, the E-field maps and supplemental information should be provided.
We added two computed maps of the power distribution in the 0th and the -1st orders resulting from the illumination of a sinusoidal gold-coated grating bounded by water (sensing medium) by a TM-polarised light beam emitting at 850 nm depending on the grating depth and the incident angle of the probing beam. The optimal depths of the gratings used in an optical switch and a conventional SPR configuration are respectively shown by a solid and a dashed white line on each map. The figures highlight the effect of the depth on the power distribution between both monitored orders and support the results presented in Figure 9. These new figures are presented in the appendix of the manuscript.

Round 2
Reviewer 2 Report
Comments and Suggestions for Authors
While the majority of the suggested changes and comments were not addressed, the work can be published as is.